# A fast impedance-based antimicrobial susceptibility test

Daniel C. Spencer[1], Teagan F. Paton[2], Kieran T. Mulroney[3], Timothy J. J. Inglis [2,3], J. Mark Sutton [4] & Hywel Morgan [1✉]

There is an urgent need to develop simple and fast antimicrobial susceptibility tests (ASTs) that allow informed prescribing of antibiotics. Here, we describe a label-free AST that can deliver results within an hour, using an actively dividing culture as starting material. The bacteria are incubated in the presence of an antibiotic for 30 min, and then approximately $10^5$ cells are analysed one-by-one with microfluidic impedance cytometry for 2–3 min. The measured electrical characteristics reflect the phenotypic response of the bacteria to the mode of action of a particular antibiotic, in a 30-minute incubation window. The results are consistent with those obtained by classical broth microdilution assays for a range of antibiotics and bacterial species.

[1] Department of Electronics and Computer Science, and Institute for Life Science, University of Southampton, Hampshire SO17 1BJ, UK. [2] Department of Microbiology, PathWest Laboratory Medicine, Nedlands, WA 6009, Australia. [3] Faculty of Health and Medical Sciences, University of Western Australia, Nedlands, WA 6009, Australia. [4] Public Health England, National Infection Service, Porton Down, Salisbury, Wiltshire SP4 0JG, UK. ✉email: hm@ecs.soton.ac.uk

Antimicrobial resistance (AMR) is a global problem resulting in a year-on-year increase in the incidence of drug resistant infections. AMR is expected to be responsible for 10 million deaths annually by 2050[1]. Excessive and otherwise inappropriate prescription of antibiotics promotes resistance; an estimated 30–50% of all antimicrobial prescriptions are unnecessary[2]. The rapid rise in multi- and pan-drug resistant infections highlights an urgent need to improve infection diagnosis and management tools to improve the stewardship of a dwindling stock of effective antibiotics. In particular, there is an immediate need for rapid tests to support evidence-based antimicrobial prescribing. Most antibiotic testing in UK hospitals is currently performed using classical culture-dependent microbiology methods that provide a susceptibility profile within 24–48 h, or longer. Consequently, antibiotics are first prescribed on a presumptive basis, without any definitive indication of their in vitro antibiotic efficacy. Unfortunately, there are no simple and fast alternative antimicrobial susceptibility tests (AST) available.

An AST can be either a genotypic or phenotypic test. Genotypic susceptibility testing classifies resistance based on the presence or absence of particular resistance genes (for example the *mecA* gene for Methicillin-resistant *Staphylococcus aureus* (MRSA)) and is only an approximation to susceptibility determination. These tests are expensive and limited to panels of known genes. Furthermore, the absence of a gene does not necessarily correlate with phenotypic susceptibility, for example carbapenem-resistant bacteria may not carry a carbapenemase gene but may have phenotypic resistance through a combination of two or more mechanisms including reduced permeability (porin switching/loss), upregulation of multidrug efflux pumps (mutations in regulator genes) and overexpression/acquisition of other non-carbapenemase β-lactamases (e.g. AmpC). This is a considerable weakness of genotypic tests given the ever-increasing range of resistance genes and the ability of bacteria to achieve phenotypic resistance through a combination of multiple mechanisms. Similarly, presence of a gene does not always equate with resistance since the gene may be weakly expressed, point mutations may affect substrate specificity, or resistance genes may be associated with other deleterious effects. Whole genome sequencing is uneconomic at present at an estimated $80 per genome based on 1-week turnaround and thus is not rapid within the clinical decision time frame[3]. In contrast, phenotypic testing evaluates the specific viability or growth response of an organism to the presence of an antibiotic and directly demonstrates whether a microbial isolate will be inhibited by the antibiotic tested. This method therefore remains the reference standard used by microbiology labs worldwide.

Phenotypic ASTs are most commonly performed using either a broth micro-dilution (BMD) or a disk diffusion assay. The BMD method provides a semi-quantitative measurement of antimicrobial susceptibility known as the minimum inhibitory concentration (MIC) for an antibiotic. Growth is measured in a range of different concentrations of antibiotic (typically a log-2 dilution series). The lowest concentration to inhibit growth visible by eye is determined to be the MIC. Although this method is used as a reference standard, it generally requires a minimum incubation of 16–24 h and sometimes longer. The internationally recognised standard for AST is MIC determination by a specific version of BMD as described in ISO 200776-1, 2006[4]. Classification of AST results into broad categories [*Susceptible* (S); *Susceptible, Increased exposure* (I) (formerly designated *Intermediate*); or *Resistant* (R)] can be made by comparing MIC results to species-specific breakpoints published by the European Committee on Antimicrobial Susceptibility Testing (EUCAST, Europe) or the Clinical & Laboratory Standards Institute (CLSI, USA). Methods for antimicrobial susceptibility testing are therefore validated against

BMD before introduction into clinical practice. Some automated AST platforms use susceptible optical readers or include metabolic probes with specialist media to provide faster results (6–8 h after initial isolation). An example of a new FDA approved imaging-based AST technology is the Accelerate Pheno system (Accelerate Diagnostics, Tuscon, AZ) that provides a sample ID using FISH and uses morpho-kinetic time-lapse imaging to provide an AST from a positive blood culture in around 6 h[5]. For a recent review on the current state of the art in AST systems see[6].

Improvements in antibiotic stewardship urgently requires the development of rapid AST, and a test that provides a susceptibility profile within a clinical shift would have a major impact on many clinical applications. A much-reduced time to result (e.g. around 1-h post-culture) would be particularly advantageous in providing information promptly enabling clinicians to expedite evidence-based prescribing. The issues and barriers that hinder the implementation of rapid tests were recently reviewed by van Belkum et al.[7], and the authors propose a roadmap for the development of new diagnostics tests.

Rapid phenotypic tests require new methods to detect changes in bacterial properties (for example morphology, membrane structure, metabolism, and cell growth) long before bacterial death occurs.

One example of a phenotypic response is the influence of the β-lactam class of antibiotics, which collectively account for 65% of worldwide consumption of antibiotics[8]. Their mode of action is through inhibition of the transpeptidase activity of penicillin-binding proteins, preventing the final stage in cross-linking of the bacterial peptidoglycan present in the cell wall. A biophysical consequence of this action is elongation or swelling of bacteria at concentrations above the MIC. In fact this effect can lead to errors in systems that rely on colorimetry or turbidometry (such as in the Vitek-2, Phoenix, MicroScan WalkAway) because larger particles may increase the light scattering used to determine cell growth[9]. β-lactam antibiotics account for 70% of US prescriptions, thus a fast and simple AST is required to accurately evaluate their activity in treatment. Particularly important are the Carbapenem class of β-lactams that resist hydrolysis by most β-lactamases and are often used as antibiotics of last resort. The World Health Organization (WHO) identified the emergence of carbapenem-resistant *Klebsiella pneumoniae* as its leading priority[10]. *K. pneumoniae* and related species are the most prominent carbapenem-resistant Enterobacteriaceae (CRE) and cause an excess hospital mortality of 27% in patients with septicaemia and pneumonia[11].

Other classes of antibiotic such as polymyxins (Colistin) cause biophysical changes in cell membrane permeability prior to cell death. Consequently, AST methods that rely on cell growth or metabolic activity do not report results in <6 h[12,13]. To address this, several rapid AST assays have been developed[13–15]. Flow cytometry has been widely proposed for rapid ASTs: antibiotic exposure leads to changes in susceptible strains of bacteria that can be measured by (label-free) light scatter and/or fluorescent viability markers[16]. However, differentiation between antimicrobial exposed and unexposed populations has proved difficult and new approaches such as adaptive multi-dimensional statistics have been developed[17]. An assay for carbapenem resistance has been developed that uses acoustic-focusing flow cytometry to deliver a rapid S/R classification together with a quantitative MIC in ~2 h from a clinical isolate[18,19]. Flentie et al.[20] introduced a novel assay that measured bacterial concentrations by binding a small-molecule amplifier to the bacterial surface. The technique delivers a phenotypic AST within 5 h for non-fastidious bacteria by measuring bacterial replication where organisms form filaments or swell in response to antibiotic exposure.

Measuring the growth rates of bacteria is an attractive means of directly determining an AST. This is usually done optically, for example Choi et al.[21] used single-cell time-lapse imaging to determine an AST in 4 h by automatically categorising morphological changes in single cells growing on a thin agarose slab in the presence of antibiotics. Baltekin et al.[22] trapped bacteria in micro-channels and monitored growth in the presence or absence of antibiotic from the change in the length of the sample in the channel. Controlled diffusion in micro-channels can be exploited to create a continuous gradient in antibiotic concentration and therefore allow determination of an MIC in a single chamber[23]. Growth rates can also be measured by detecting changes in the mass of bacteria using resonant cantilevers[24]. The system was integrated within a microfluidic channel, bacteria were captured with antibodies and the response determined in a short time window (30 min)[25]. Rapid methods based on electrochemical labels have also been reported. Metabolically active bacteria were detected through the reduction of resazurin giving an antibiotic susceptibility profile in 1 h[26]. Resazurin has also been used as an optical probe to rapidly determine the phenotypic susceptibility in nano-litre volume arrays[27]. The growth rate kinetics of exposed and control samples were compared, and an antibiotic profile obtained in 4–5 h.

In this paper we describe a rapid label-free phenotypic AST that delivers a resistance profile in as little as 30 min. The technique which we call impedance-based Fast Antimicrobial Susceptibility Test (iFAST) measures changes in the electrical and morphological properties of many thousands of single organisms at high throughput using microfluidic impedance cytometry. In order to align the test with standard microbiology protocols, an inoculum is first taken from an overnight bacterial culture. This is resuspended in growth medium for 30 min and then incubated with an antibiotic for a further 30 min. Approximately $10^5$ bacteria are measured in a time window of 2–3 min to determine a response profile (Fig. 1). The utility of the method was first demonstrated by measuring the MIC for carbapenem-resistant *K. pneumoniae*. Rapid analysis at the Susceptible/Resistant clinical breakpoints (EUCAST v10) was demonstrated for carbapenems against *Escherichia coli*, *Acinetobacter baumannii*, and *Pseudomonas aeruginosa*. We also show that the technology is capable of identifying resistance profiles for a wide range of antibiotics with different modes of actions (Colistin, Aminoglycosides (Gentamicin), Fluoroquinolones (Ciprofloxacin), Cephalosporins (Ceftazidime), and antibiotic/inhibitor combinations (Co-amoxiclav) against Gram-negative organisms, and Cefoxitin against *S. aureus* (MRSA)). Together, these organisms contribute the greatest number of directly attributable deaths in Europe;[28] they are also included in the WHO Priority drug resistant pathogens[29].

## Results

**Measurement principle**. iFAST measures changes in the biophysical properties of bacteria after exposure to antibiotics measured by microfluidic impedance cytometry, a well-established method that has been widely used for label-free characterisation of mammalian cells[30–32]. The technique measures the electrical properties of single particles as they flow between microelectrodes within a microfluidic chip (Fig. 1a). The electrodes are driven by an AC signal of several frequencies and when a cell flows along the channel it perturbs the AC current; the measured change is the impedance for the individual particle[30,33]. Despite widespread use of the technique, measurement of micron-sized particles has proved challenging requiring specialised electronics[34], complex microfluidic approaches[35,36], or shallow channels (differentiation of Gram-negative from Gram positive bacteria)[37].

In this work we show that accurate analysis of bacterial properties by impedance is made possible using a system with considerably improved sensitivity (Signal to Noise Ratio, SNR) allowing rapid measurement of micron-sized particles in a channel with a large cross section (~20 μm × 40 μm) with a limit of detection of ~400-nm radius (see Supplementary Fig. 1 for details). The principle of measurement is similar to a conventional impedance cytometer device where cells suspended in an electrolyte flow along the channel one-by-one through two pairs of electrodes (Fig. 1a). In a conventional cytometer system two pairs of electrodes measure a differential signal. The new electrode arrangement (Fig. 1a and Supplementary Fig. 1) uses two pairs of electrodes in each arm of a differential circuit thus reducing the baseline current (no cell) in each of the transimpedance amplifiers, enabling higher amplifier gain, and higher SNR. This method enables small particles such as microorganisms to be characterised at high speed (up to 1000/s) in high conductivity media in a large channel with minimal risk clogging by cells and debris (and low fluid back pressure). It thus provides a new way of characterising subtle biophysical changes in bacteria, enabling the effects of antibiotic exposure to be measured after a very short time window. Micron-sized polystyrene beads are added to every sample as reference particles. These beads have well defined electrical properties (and size), and are used to eliminate device to device variation and non-linearities in the measurement electronics[38].

The electrical properties of a cell are generally characterised using a simple equivalent electrical model. An example is shown in Fig. 1b[30], modified to include the double membrane of a Gram-negative bacterial cell. At low AC frequencies, the bacteria behave as insulators so that the impedance signal is proportional to cell volume. At higher frequencies other effects influence the impedance signal, particularly changes in cell wall and cell membrane. These effects are shown in Fig. 1b where a simulated spectrum of the Real part of the impedance signal (differential current) vs. frequency shows the frequency windows where changes in cell properties are apparent. At low frequencies the measurement principle is similar to the Coulter counter; the impedance signal directly correlates with cell volume. Thus, changes in for example cell length or filamentation are measured at low frequencies (although the high frequency signals are also modulated). The low frequency impedance is also influenced by cell wall conductivity; a cell with an electrically leaky membrane is no longer a perfect insulator and its apparent volume will thus decrease. Changes in cytoplasmic conductivity only affect the high frequency part of the spectrum, whereas changes in membrane or cell wall capacitance (permittivity) are observed in the mid-frequency range of the spectrum. In other words, several different phenotypic responses can influence the measured signal, depending on the applied frequency. To factor out the influence of cell size on the high frequency impedance measurement, the ratio of high-to-low frequency impedance is typically reported as the "electrical opacity". The net contribution of each of these separate elements to the total signal provides an electrical fingerprint for an organism.

β-lactam antibiotics specifically target the bacterial cell wall, interfering with cell division, and the maintenance of cell wall synthesis causing filamentation or spheroplast formation[9]. These phenotypic changes in size and/or cell wall lead to changes in the electrical properties as shown in the impedance scatter plot of Fig. 1c where data for *K. pneumoniae* before and after exposure to Meropenem are plotted. A measurable shift is observed in the electrical parameters of the population along both axes (refer to red contour). Antibiotic exposed cells increase approximately threefold in volume (mean diameter increase by 50%, from 1.8 to 2.6 μm). Antibiotic-induced changes in the cell wall also lead to a

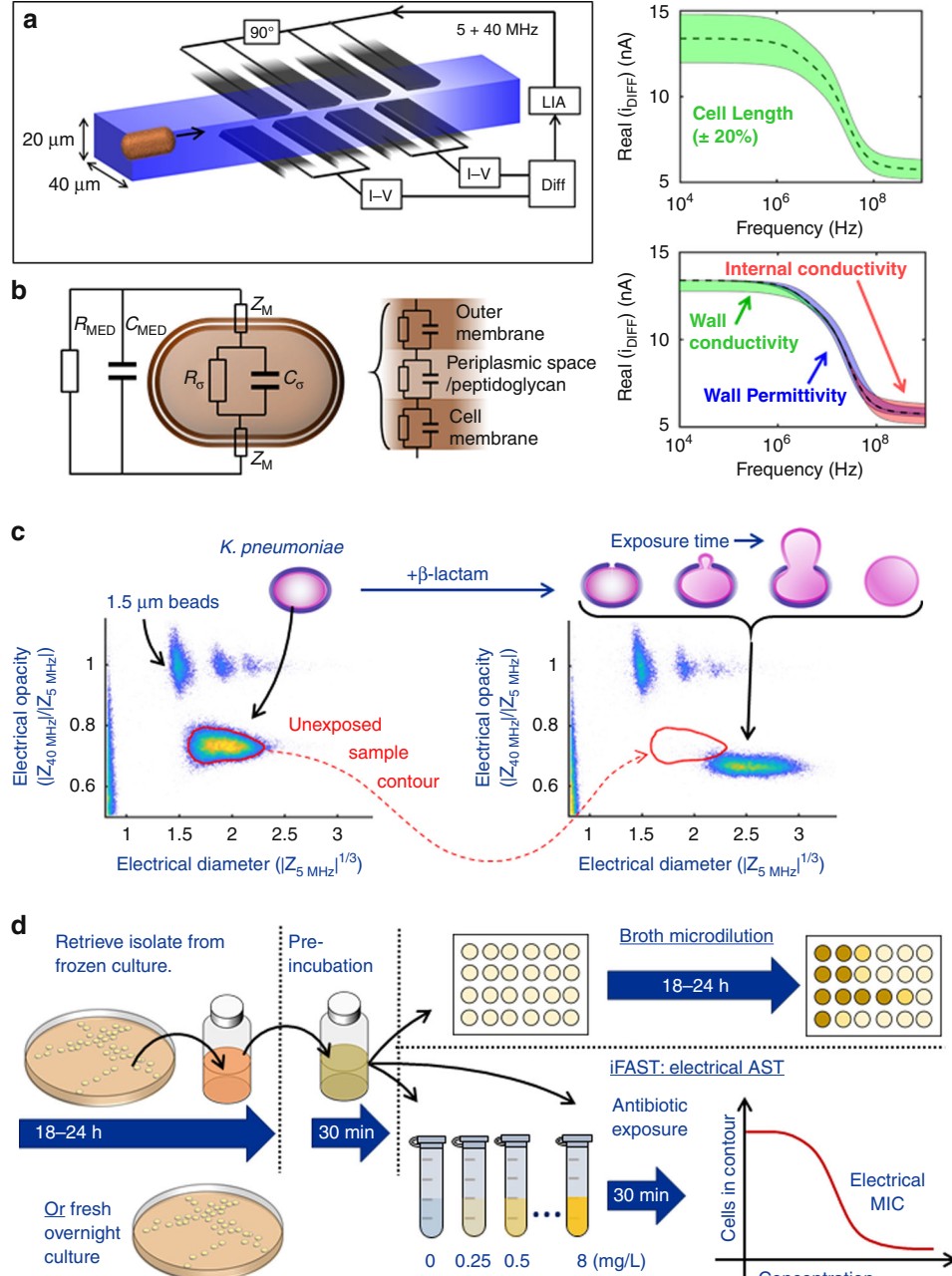

**Fig. 1 Principle of rapid impedance-based antimicrobial susceptibility testing. a** Multi-electrode microfluidic impedance chip. Cells flow one-by-one between sets of electrodes and are measured simultaneously at two frequencies using a lock-in-amplifier (LIA). **b** Equivalent electrical equivalent circuit model for a Gram-negative bacteria, and a simulated spectrum of the Real part of the impedance vs. frequency highlighting frequency windows where changes in cell properties become apparent. **c** Impedance scatter plot of bacteria (*K. pneumoniae*, 10,000 events) together with 1.5-μm diameter polystyrene beads (with doublets and triplets) that are used as reference particles. The x-axis is the cube root of the impedance (proportional to diameter) measured at a frequency of 5 MHz. The y-axis is the electrical opacity, a measure of membrane/cell wall properties normalised to cell volume. This is measured at 40 MHz where the electrical properties of the cell wall and membrane are most apparent (see **b**). Two data sets are pre- and post-exposure to Meropenem at the clinical breakpoint for 30 min at 37 °C. In the scatter plot, the red contour defines the initial cell population. The diagram illustrates the change in cell properties following exposure to a β-lactam antibiotic as the cell wall breaks down (reduction in opacity) and the bacteria swell (increase in volume). **d** Experimental methodology for the impedance-based Fast Antimicrobial Susceptibility Test (iFAST). An actively dividing culture is prepared and incubated for 30 min with antibiotics. Polystyrene beads are added and the sample is measured for 3 min to determine the electrical MIC (see Supplementary information for further details).

reduction in electrical opacity, (approximately in inverse propor-tion to cell capacitance) reflecting structural changes in the cell wall. This change in impedance of bacteria occurs as early as 10 min post-exposure and continues for over 30 min (typical doubling time). This effect is shown by the time series in Fig. 2.

Changes in electrical phenotype are also observed when cells are exposed to other classes of antibiotics. Certain antibiotics (e.g. Colistin) directly alter membrane properties which is reflected in changes in the electrical cell size, whilst others (ciprofloxacin or gentamicin) inhibit DNA gyrase and

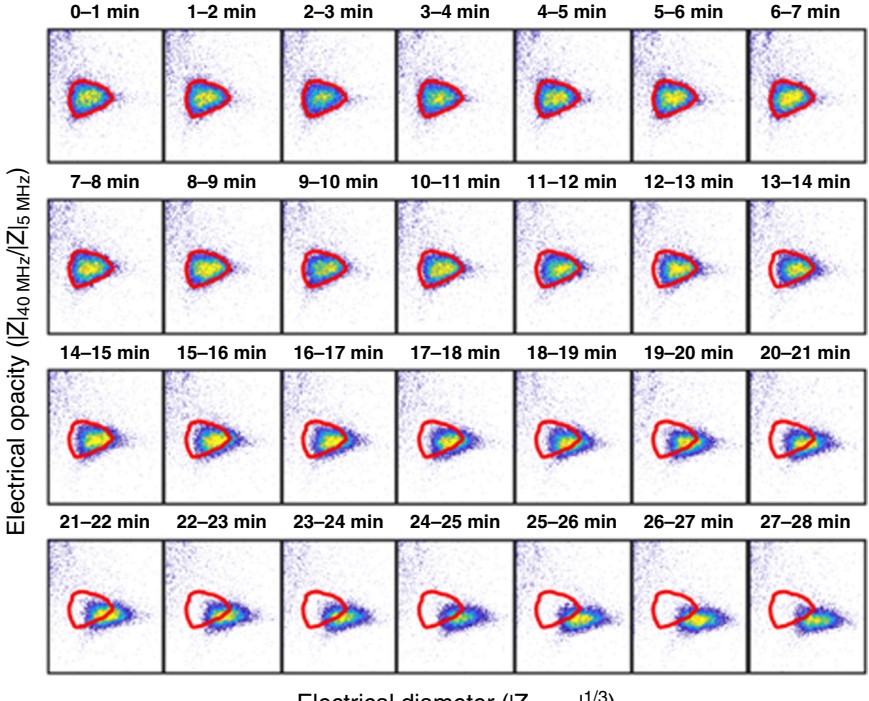

**Fig. 2 Time course of changes in electrical properties of *K. pneumoniae* following β-lactam exposure.** A sample of a Meropenem-susceptible strain of *K. pneumoniae* was exposed to Meropenem at 2 mg/L. The sample was maintained at a temperature of 37 °C and measured continuously for 30 min. The data was segmented into 1 min intervals and plotted as a series of scatter plots. Cells measured in the first minute were used to define the reference contour shown in red. Note that the x- and y-axis limits are identical for all figures (not shown for clarity) and are 1.2–3.0 (x-axis) and 0.4–0.9 (y-axis). Source data are provided as a Source Data file.

protein translation respectively releasing toxic intermediates following inhibition of essential cellular processes and ultimately lead to cell death.

The principle of iFAST is shown in Fig. 1d and is designed to mirror a typical microbiology lab workflow. After an overnight culture (as for a standard AST) bacterial cultures are incubated for 30 min at 37 °C to ensure they are actively dividing. The cultures are then exposed to antibiotics of various classes and concentrations for a further 30 min before measurement (3 min) using impedance cytometry. The MIC of an antibiotic is determined by measuring the electrical response of the same isolate exposed to different concentrations of antibiotic. Microbiology laboratories generally report strains as *susceptible*; *susceptible, increased exposure* or *resistant* by breakpoint analysis using interpretive criteria. iFAST can also distinguish sensitivity and resistance at fixed breakpoint concentrations for different bacterial species and antibiotic classes.

**Minimum inhibitory concentration (MIC): carbapenem.** The electrical MIC obtained using iFAST was compared with the MIC determined using standard BMD for three different strains of *K. pneumoniae* [*susceptible* (strain 18397); *susceptible, increased exposure* (strain KS11); or *resistant* (strain K14)] exposed to six different Meropenem concentrations, measured according to protocol 1 (Supplementary Fig. 2a). Figure 3a shows a set of scatter plots of electrical opacity vs. electrical diameter for these isolates. For the *susceptible* strain (18397) changes in electrical properties are observed even at the lowest concentration of antibiotic, whilst K14 (*resistant*) shows no changes at up to 8 mg/L. The MIC of the strains determined by BMD (in triplicate) was K14 = 128 mg/L, KS11 = 8 mg/L, and 18397 < 0.25 mg/L. Figure 3b shows the electrical MIC for ten different strains of *K. pneumonia* that have a

range of different MICs (see Supplementary Table 1 for details of strains). The data is plotted as the % cells within a contour (or gate) defined by the unexposed population vs. antibiotic concentration (for three biological replicates). Qualitatively the data shows that there are three different "classes" of response. The three resistant strains (red) all demonstrate no change in the exposed vs. unexposed gate. The five *susceptible* strains (blue) all demonstrate a large change in the scatter plot and absolute cell count for the lowest concentration of antibiotic (0.25 mg/L). The *susceptible, increased exposure* strains (orange) fall to >50% cell count within the gate at an antibiotic concentration >2 mg/L. The accepted definition for a BMD MIC is inhibition of growth visible by eye, but there is no equivalent EUCAST standard for fast MIC tests. Assuming a doubling time of 30 min, a bacteriostatic agent with no biophysical changes would approximately halve the number of cells compared with a control gate. If the MIC is calculated at an assumed threshold of 50%, all strains have a MIC within a single twofold dilution of the BMD results.

Exposure of actively dividing Meropenem-susceptible isolates to inhibitory concentrations of the drug has been demonstrated to produce a range of cellular morphotypes; cells elongate, swell, balloon, and eventually proceed to complete cell lysis as they become compromised. The impedance measurement data shown here is consistent with this range of changes but measured in very short time frames. As the scatterplots show, there is a population shift out of the original unexposed contour; in this case almost 100% of the cells migrate within the 30-min exposure. At high antibiotic concentration of 8 mg/L, small numbers of cells remain in the original gate for both 18397 and KS11. These could be a population of non-viable cells that are electrically leaky (18397), or resistant cells (KS11), or may simply reflect the time taken by Meropenem to kill all of the cells in the population given its mode of action.

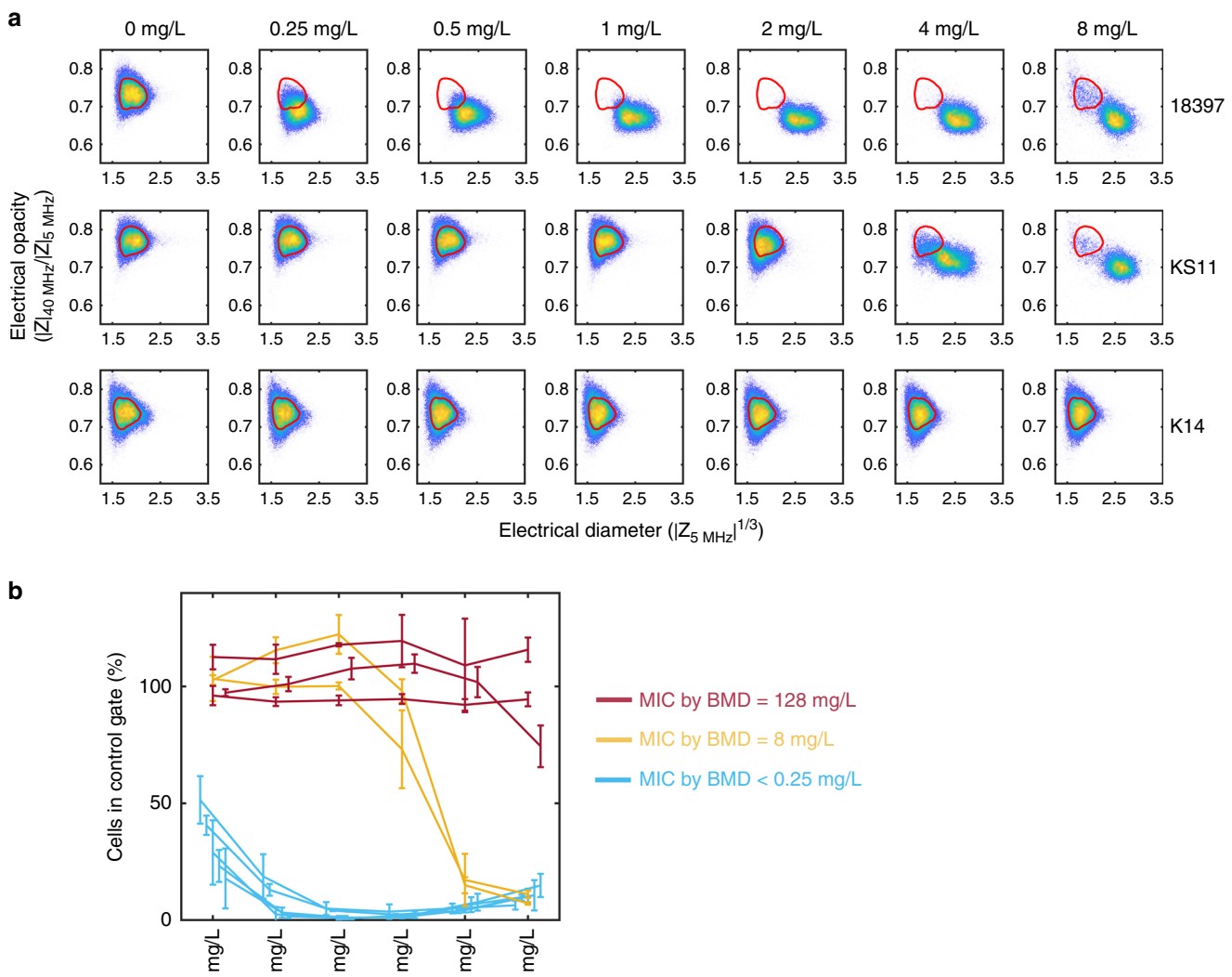

**Fig. 3 Electrical minimum inhibitory concentration (MIC). a** Scatter plots showing electrical size vs. electrical opacity for three different strains of *K. pneumoniae* exposed to different concentrations of Meropenem ranging from 0 to 8 mg/L (see Supplementary Fig. 2, protocol 1) Top row: 18397 (*susceptible*), middle row: KS11 (*susceptible, increased exposure*), bottom row: K14 (*resistant*). No changes are observed across all concentrations for K14, whilst a small shift in population is observed for 18397 even at 0.25 mg/L. Broth micro-dilution (BMD) was used (same samples) to measure the MIC and to classify the strains (red = 128 mg/L, orange = 8 mg/L, and blue < 0.25 mg/L). **b** Electrical MIC for ten different strains of *K. pneumoniae* that have a range of different MICs (see Supplementary Table 1). The data is plotted as the % cells in a contour (or gate) defined by the unexposed population vs. antibiotic concentration for three biological replicates (mean ± SD). Source data are provided as a Source Data file.

**Breakpoint analysis**. Clinical breakpoints are based on the epidemiological cut-off values taken from bacterial culture collections, and define antibiotic concentrations that enable interpretation of the results of MIC tests to classify bacterial isolates as *susceptible (S); susceptible, increased exposure (I);* or *resistant (R)* to that antibiotic when used therapeutically. Breakpoints reflect drug potency against a population of potential pathogens, the pharmacokinetics/pharmacodynamics of the antibiotic and the dosing regimens that may be achievable in the clinic. For example, isolates of *Enterobacteriaceae* with an MIC of 2 mg/L Meropenem or lower are defined by EUCAST as *susceptible*, and an MIC greater than 8 mg/L is defined as *resistant*. A MIC > 2 mg/L, but no more than 8 mg/L is in an intermediate category that may require an increased Meropenem dose for some infections caused by this bacterial isolate. We tested the utility of the iFAST technology to rapidly measure the breakpoint for different antibiotics and priority pathogens (see Supplementary Table 1) to see if it could correctly classify strains as

susceptible or resistant, using protocol 2 (Supplementary Fig. 2b). In these experiments we measured growth, cell volume and membrane biophysical changes after incubation with Meropenem at the clinical breakpoints (2 mg/L; susceptible and 16 mg/L; resistant). The data was quantified by measuring the number of cells in the unexposed contour after 30 min incubation with Meropenem (Fig. 4). Figure 4a summarises the results for strains measured at the *S/I* boundary (inhibition of growth at 2 mg/L indicates a susceptible strain), whilst Fig. 4b summarises measurements where strains were exposed to a higher concentration (16 mg/L). This concentration was selected because growth at >8 mg/L indicates resistance according to EUCAST guidelines. The bars are coloured according to standard BMD data with red indicating resistant strains and blue for susceptible strains. In all cases, the difference between resistant and susceptible strains is statistically significant as indicated in the figure.

The orange bar labelled KP CNCR (carbapenemase negative carbapenem resistant) is an isolate of *K. pneumoniae* that has

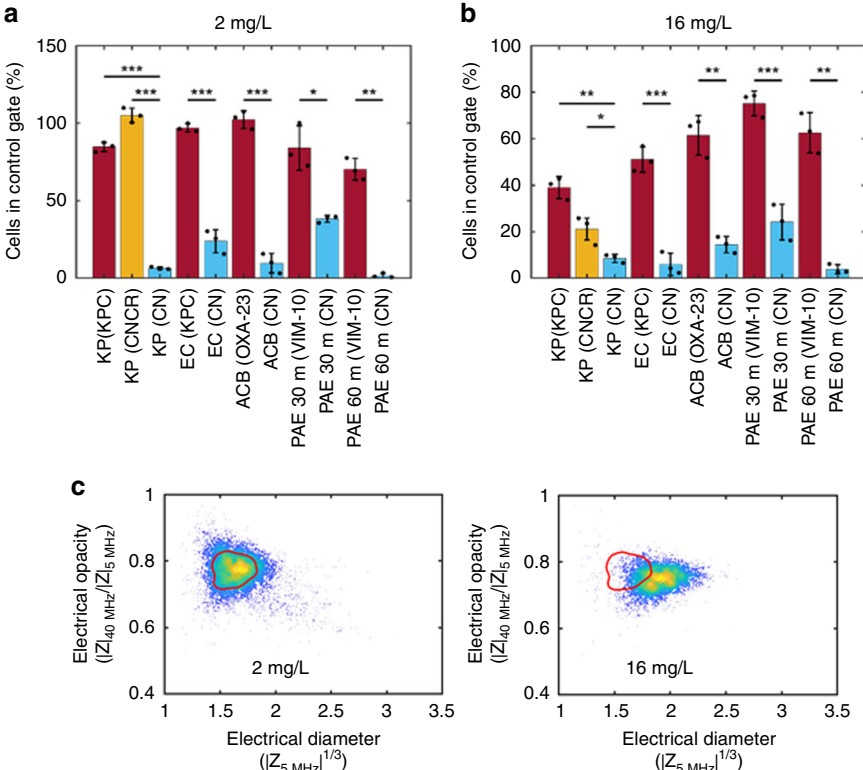

**Fig. 4 Electrical breakpoint analysis for Meropenem.** Eleven different strains of bacteria analysed by impedance cytometry after exposure to Meropenem at the clinical breakpoint. (KP *K pneumoniae*, EC *E. coli*, ACB *A. baumannii*, PAE *P. aeruginosa*) (see Supplementary Fig. 2, protocol 2). **a, b** Each bar is the percentage of cells in the unexposed contour (or gate) after exposure to antibiotics at the clinical breakpoint. The *susceptible/susceptible, increased exposure* (S/I) and *susceptible/resistant* (S/R) boundaries are 2 and 16 mg/L, respectively. The bars show the mean ± SD for three biological replicates (*$p < 0.05$; **$p < 0.01$; ***$p < 0.001$) with $p$ values obtained using the Student's $t$ test for independent samples (one tailed). Red bars indicate resistant strains and blue bars indicate susceptible strains, as determined by broth micro-dilution. The orange bar (CNCR) is a carbapenemase negative strain that is carbapenem resistant. $P$ values (from left to right) are as follows $1.04 \times 10^{-4}$, $2.66 \times 10^{-4}$, $7.14 \times 10^{-4}$, $2.46 \times 10^{-5}$, $1.43 \times 10^{-2}$, $1.15 \times 10^{-3}$, $1.97 \times 10^{-3}$, $3.00 \times 10^{-2}$, $2.19 \times 10^{-4}$, $2.40 \times 10^{-3}$, $6.13 \times 10^{-4}$, $2.63 \times 10^{-3}$. **c** Scatter plots for a CNCR strain of *K. pneumoniae* after exposure to Meropenem for 30 min at the S/NS breakpoint (2 mg/L) and the R/NR breakpoint (16 mg/L) concentrations. Source data are provided as a Source Data file.

no carbapenemases, but is resistant to carbapenems under standard testing, probably due to a combination of porin loss and upregulated AmpC expression. At higher concentrations of Meropenem, a decrease in cell count and change in biophysical properties was observed, but less than in the susceptible isolate (Fig. 4c). *E. coli* expressing KPC-2 is clearly differentiated from *E. coli* carbapenemase negative (compare bars 4 and 5 in Fig. 4a, b). Carbapenem-resistant *A. baumannii* expressing Oxa-23 showed a similar profile to that seen for *K. pneumoniae*, with both a change in opacity and electrical radius, and could be clearly differentiated from a susceptible isolate (bar 6 and 7). As *P. aeruginosa* has a slower growth rate compared to other strains tested in this study, the same samples were measured after 30 and 60 min antibiotic exposure (see bars 8–11). Differences in profile between susceptible and resistant strains are still apparent after 30 min especially at the higher concentrations, despite the slower growth rate. The longer incubation (1 h) improves discrimination between the Meropenem-susceptible and Meropenem-resistant isolate at the lower concentration of Meropenem (Fig. 4a, bars 10 and 11). In all cases the differences between resistant and susceptible isolates were shown to be statistically significant, across triplicate experiments.

While β-lactam agents are the most widely utilised antibiotics, other front-line antibiotics play a considerable role in treating infections but have different modes of action. The utility of iFAST to determine susceptibility, independent of antibiotic mechanism, was studied with a range of species-antibiotic combinations,

chosen as having the highest morbidity/mortality as outlined in a recent pan-European study[28]. This work examined the impact of antibiotic-resistant bacteria and identified increased incidence of infection with antibiotic-resistant bacteria combinations taken from the European Antimicrobial Resistance Surveillance Network (EARS-Net) 2015[28]. These organisms were also included in the European Centre for Disease Prevention and Control (ECDC) point prevalence survey of health-care-associated infections and antimicrobial use (2011–2012), and in the list of EU antibiotic resistance policy indicators published as a joint scientific opinion by the ECDC, European Food Safety Authority, and European Medicines Agency[28]. The outcome of the study identified a number of antibiotic–pathogen combinations that have the largest impact measured in DALYs (see Fig. 1 in ref. [28]). Despite a low incidence, carbapenem-resistant *K pneumoniae* had a high burden of disease because of its high attributable mortality. Other high impact combinations include carbapenem resistant *P. aeruginosa*, carbapenem-resistant *Acinetobacter spp*, third-generation cephalosporin-resistant *E. coli*, third-generation cephalosporin-resistant *K. pneumoniae*, Colistin-resistant *K. pneumoniae*, and MRSA. We also measured aminoglycoside (gentamicin) and a β-lactam with β-lactamase inhibitor (e.g. co-amoxiclav), a commonly used front-line treatment.

To validate the broad utility of the impedance technology, we set out to demonstrate the ability of the approach to differentiate between resistant and susceptible isolates treated with different antibiotic classes on the basis of breakpoint determination. In

each experiment we looked at a resistant and a susceptible isolate, focussing on the high burden pathogen–antibiotic combinations identified above. As shown in Fig. 5, it was possible to clearly differentiate between the susceptible and resistant strains in all examples tested, with statistically significant differences. Intriguingly, the pattern of migration out of the control contour differed between different antibiotic classes. For β-lactam type antibiotics, ceftazidime, and co-amoxiclav, there was a clear increase in the electrical diameter of the susceptible population, similar to that seen with carbapenems (Fig. 3) although this was greater in ceftazidime-treated *K. pneumoniae* than for similarly treated *E. coli*. Interestingly, only co-amoxiclav treated *K. pneumoniae* showed the decrease in electrical opacity that had previously been seen with Meropenem treatment. Cefoxitin, used here as a surrogate for methicillin in line with EUCAST testing protocols, gave a very different response in *S. aureus*. No changes were seen in the population for MRSA (constitutive MecA expressing strain), but a reduction in the electrical diameter was observed for MSSA after exposure to antibiotic. This metric allows rapid discrimination of resistant from susceptible clinical isolates. The electrical opacity for both MRSA and MSSA did not change (see Supplementary Fig. 3) but plotting the electrical phase against diameter (Fig. 5) enhances discrimination for the MSSA isolate. The change in phase may reflect the different role of peptidoglycan in Gram-positive compared to Gram-negative bacteria which translates into a different impedance spectrum.

Ciprofloxacin, a fluoroquinolone, induced a small increase in electrical size in a susceptible population compared to the resistant strain. Ciprofloxacin inhibits DNA gyrase and leads to the accumulation of DNA fragments and leakage from the cell. Binding of the drug to the gyrase causes double-strand DNA breaks which lead to suppressed cell division and a change in the aspect ratio[39,40] although these effects take several hours to fully develop. This is consistent with observations in Fig. 5, where the mean volume of the population approximately doubles after 30 min exposure.

For Colistin and gentamicin, there was a small shift in the cell populations in the resistant isolates, possibly reflecting small changes in the electrical properties of the bacterial membrane upon interaction with these cationic compounds. Despite the small shift in the mean position of the resistant population relative to the contour, much larger differences were observed with the susceptible isolates. A reduction in the electrical diameter is observed for both antibiotics, together with a reduction in the total cell count in the case of gentamicin, as shown by the decrease in the density of the scatter plot for the exposed population. Gentamicin is a widely use aminoglycoside and suppresses protein synthesis by binding to the ribosome. It also permeabilises the membrane due to its cationic characteristics at physiological pH[41] which correlates with the observed small changes in electrical properties of the resistant population (increased opacity and reduction in apparent volume). For the susceptible cells, the total count is markedly reduced after 30 min and a large decrease in apparent cell size is seen, consistent with an increase in the permeability of the membrane. A similar trend following exposure to Colistin is observed. This cationic polymyxin interacts with the outer membrane leading to deformation, pore formation and leakage. It permeabilises the cytoplasmic membrane, ultimately leading to cell lysis and death. The observed reduction in the measured electrical cell volume correlates with an increase in membrane permeability[42]. In both cases, cell volume did not increase consistent with the absence of any filamentation[43].

Each antibiotic–bacteria combination test was repeated three times and the data summarised in the bar chart. For all antibiotics tested (except gentamicin and Colistin) the numbers of cells in

the exposed resistant population matches with the unexposed population as shown by the red bars at ~100%. For Colistin and gentamicin, the small shift in the resistant populations seen in the scatter plot mean that the cell count in the unexposed contour is reduced. Nevertheless, the susceptible strains are all statistically different ($p = 0.01$).

## Discussion

This work demonstrated an ultra-rapid AST that measures the electrical properties of thousands of single bacteria to determine a susceptibility profile in a very short time window. The assay is label-free and extremely simple; involving exposure of a bacterial suspension to antibiotics, incubation (at 37 °C), dilution and measurement. This technique mirrors the reference standard BMD assay in terms of both the phenotypic rationale for the measurement and the demonstrable relationship between MIC measurement methods, but is much quicker. Continuous direct measurements of bacteria directly in media containing antibiotics is also possible by monitoring growth and biophysical changes in real-time (Fig. 2). This provides added value as a research tool to understand the responses of bacterial populations to antibiotics at the single-cell level.

Phenotypic analysis is the agreed standard for antimicrobial susceptibility testing, largely because presence or absence of a resistance gene does not perfectly correspond to susceptibility to an antibiotic. An example of this is carbapenemase-negative, but carbapenem resistant (CNCR) *Enterobacteriaceae*; the summary data in Fig. 4 demonstrates that iFAST can detect CNCR strains. Of note, the CNCR cells show a phenotypic response to Meropenem at high antibiotic concentrations, compared to cells with KPC which is a very efficient carbapenemase. iFAST technology has been demonstrated for both rapid MIC and breakpoint analysis. For antimicrobial stewardship, breakpoint analysis provides the bulk of data used by prescribing physicians. The MIC determination is used only where an understanding of the level of susceptibility of an isolate is required (for example in surveillance, epidemiology, or mechanistic studies).

Optical flow cytometry has been widely utilised to monitor bacterial growth by a combination of cell size (determined from optical scatter) and fluorescent dyes (e.g. to measure viability)[16–18]. In comparison, impedance cytometry is a label-free method that directly measures both cell volume (size) and other phenotypic changes that are reflected in the electrical signature. In iFAST, we observed changes in the electrical properties of cells due to the action of antibiotics such as β-lactams, whereas only small changes in optical scatter signal are observed using flow cytometry[16,44,45]. Unlike electrical techniques, optical methods indirectly determine cell volume from light scattering and the signal can be influenced by cell refractive index, shape, and orientation[46], and by debris in the suspension. To overcome this, a combination of optical scatter and fluorescence dyes are used as a dual trigger[47]. However, the use of fluorescent dyes precludes real-time measurements, because many dyes intercalate with DNA and inhibit growth. Dyes are often expensive and demand more sample preparation steps, including washing and centrifugation. Impedance analysis can perform continuous real-time measurement (Fig. 2) and easily discriminates debris from cells without a "label" because the electrical properties of cells are distinct. In this work we have demonstrated that the changes observed in the electrical properties of bacteria following antibiotic exposure are easy to resolve without recourse to complex statistical analysis[17].

Overall, the iFAST approach shows utility for the rapid detection of antibiotic susceptibility across a range of clinically

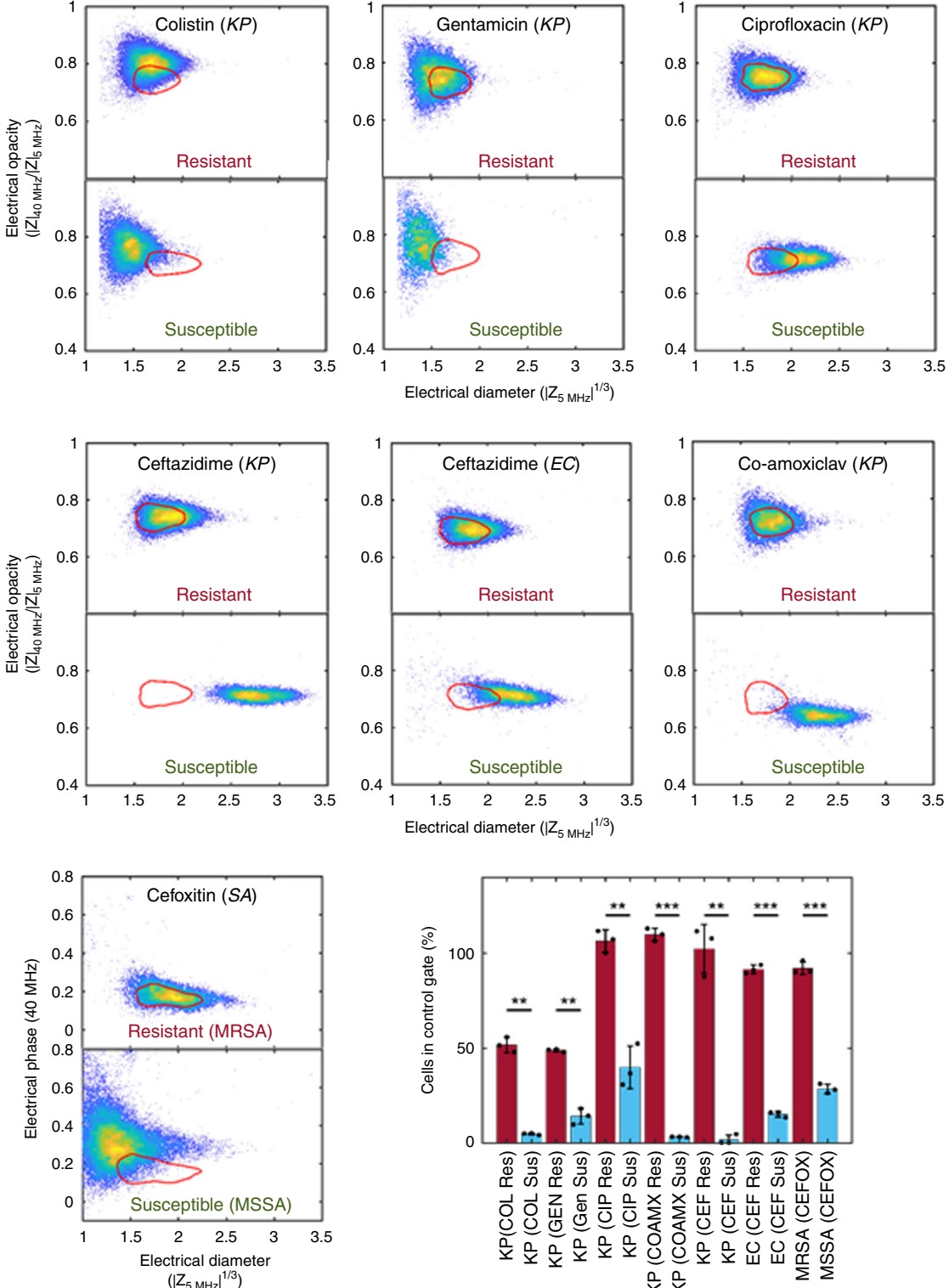

**Fig. 5 Breakpoint analysis for different antibiotic mechanisms.** Scatter plots for susceptible and resistant bacteria after exposure to different antibiotics (see inset labels) at the *S/R* clinical breakpoint. The figure shows *K. pneumoniae* (KP), *E. coli* (EC), and *S. aureus* (SA) exposed to antibiotics which have different modes of action. The bar chart shows the percentage of cells inside the unexposed contour (mean ± SD for three biological repeats, red bar: resistant strain, blue bar susceptible strain as determined by broth micro-dilution) *$p < 0.05$; **$p < 0.01$; ***$p < 0.001$, with $p$ values obtained using the Student's $t$ test for independent samples (one tailed), from left to right as follows: $1.04 \times 10^{-3}$, $1.83 \times 10^{-3}$, $1.36 \times 10^{-3}$, $1.53 \times 10^{-3}$, $2.06 \times 10^{-3}$, $1.96 \times 10^{-6}$, $1.33 \times 10^{-5}$. Source data are provided as a Source Data file.

important pathogen–antibiotic combinations. The simplicity of the measurement technique suggests that the method is suitable for a new generation of rapid tests for the clinical laboratory. The ability of the technique to measure at the single-cell level, provides considerable benefit to resolve largely unseen responses in bacterial populations being treated, such as being able to monitor sub-populations that may be resistant or tolerant of the antibiotic and the emergence of resistance in near real-time. This provides a direct biophysical measure of the properties of these sub-populations which may help us understand these complex phenomena and pave the way towards the development of improved therapies.

## Methods

**Assay protocols**. Two different protocols were used in evaluating antimicrobial susceptibility[48], summarised in Supplementary Fig. 2. In Protocol 1 (Supplementary Fig. 2a), the iFAST electrical MIC protocol was designed to mirror a classical micro-dilution assay. A colony was picked from a plate and incubated overnight in Tryptic Soy Broth to stationary phase. An aliquot of this culture was diluted into Cation Adjusted Mueller Hinton Broth (MHB) to an approximate concentration of $5 \times 10^5$ cells/mL. The bacterial concentration was determined by measurement with the microfluidic impedance cytometer. The sample was then incubated at 37 °C for 30 min to obtain an actively dividing culture. Aliquots (950 μL) were added to 7 pre-warmed test tubes each containing 50 μL MHB and Meropenem at a final antibiotic concentration of 0, 0.25, 0.5, 1, 2, 4, or 8 mg/L. The tubes were incubated for 30 min at 37 °C, cells washed once in Hanks Balanced Salt Solution (HBSS) then diluted 1:10 in HBSS. 1.5-μm diameter polystyrene beads (reference particles, Polysciences) were added to each aliquot (@10^4/mL). Finally, each sample was loaded into a syringe and measured by pumping it through the impedance chip at a flow rate of 30 μL/min for 3 min. In parallel an aliquot of the actively dividing culture was taken and used for a standard BMD assay.

Protocol 2 (Supplementary Fig. 2b) measures the phenotypic response at the antibiotic breakpoint with the concentration(s) of antibiotics fixed at a pre-defined concentration. For this assay, three colonies were picked from a plate and added to 3 mL of MHB. The sample was vortexed to re-suspend the bacteria and then diluted to a concentration of $5 \times 10^5$/mL in MHB. The sample was incubated for 30 min at 37 °C to obtain an actively dividing culture. Aliquots of 500 μL were added to test tubes containing a pre-warmed volume (500 μL) of MHB, each with a final antibiotic concentration at the clinical breakpoint according to EUCAST guidelines: 2 mg/L and 16 mg/L for Meropenem (S/I, and S/R boundary), 1 mg/L for ciprofloxacin, 8 mg/L for gentamicin, 4 mg/L for Colistin, 8 mg/L for ceftazidime, along with a control (no antibiotic). Each tube was incubated for 30 min at 37 °C (antibiotic exposure), then the sample diluted 1:10 in HBSS. 1.5-μm diameter beads were added and the sample measured for 2 min at 30 μl/min in the impedance micro-cytometer.

**Impedance micro-cytometer**. Microfluidic chips were fabricated using photolithography and wafer bonding. Briefly, metal (Pt) electrodes were patterned onto two 6-inch glass substrates (200 nm Pt and 10 nm Ti patterned by ion beam milling). Channels (20-μm deep) were made by patterning SU8 onto one wafer. The second wafer was bonded to the first wafer by vacuum bonding at 180 °C, 10 kN for 6 h. Channels had cross sectional dimensions of 20 μm × 40 μm and electrodes were 30-μm wide with 10-μm gaps. Fluidic connections were made using custom 3-D printed acrylic interconnects that utilised 1.6 mm OD 0.5 mm ID Teflon tubing with Teflon gripper ferrules. Bacterial suspensions were loaded into a 1 mL syringe and pushed through the impedance cytometer chips with a Harvard Instruments syringe pump at 30 μL/min. The impedance signal of each cell was measured using a Zurich Instruments impedance scope (HF2IS) and custom PCB front end amplifier board connected to the glass micro-cytometer using Samtech SEI series connectors. Two frequencies (5 and 40 MHz) were applied simultaneously to the electrodes. A signal of 4 V was used and the differential current sampled at 230k samples per second.

**Data analysis**. The impedance data signals were processed using custom software written in MATLAB. The impedance of each particle was determined from the peak signal amplitude for each applied frequency using convolution. The mean signal of the 1.5-μm beads was determined automatically in each experiment by searching within a pre-defined gate/contour. The opacity-cell size scatter plot was normalised by a single linear multiplier for each axis to ensure the mean of the beads is at opacity = 1 and diameter = 1.5 μm. Several parameters can be examined to determine susceptibility—e.g. size change due to β-lactams, or change in electrical opacity due to cell wall changes, or a decrease in growth rate. These are all captured by comparing the unexposed sample to the exposed sample. A contour is defined automatically around the population of cells in the aliquot not exposed to antibiotic. This is termed the unexposed contour and is calculated automatically using a density plot of the cells to include 50% of the cell population. A decrease in growth rate (indicating a

susceptible strain) results in fewer cells in the control sample. Equally, a change in biophysical properties (susceptible strain) but continued growth moves some or all of the exposed sample outside the control gate and thus leaves fewer cells in the control gate. Statistical comparisons between susceptible and resistant or untreated populations were carried out using a one tailed Student's $t$ test with $p$ value ranges given in the figure captions.

**Reporting summary**. Further information on research design is available in the Nature Research Reporting Summary linked to this article.

## Data availability
The data supporting this study are openly available from the University of Southampton repository at https://doi.org/10.5258/SOTON/D1405 which contains the source data underlying Figs. 2, 3, 4, 5 and Supplementary Fig. 3. Source data are provided with this paper.

## Code availability
The impedance data was collected using ZiControl (version 19.05, Zurich Instruments and is freely available at https://www.zhinst.com). Data analysis scripts were developed in MATALB (version 2019a, Mathworks) and are available from the corresponding author on reasonable request. Statistical analysis was performed using standard functions available in MATLAB.

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

## Acknowledgements

We would like to thank Ying Tran for fabricating the micro-cytometer chips. H.M. would like to thank the Royal Society for funding. T.J.J.I and D.C.S. acknowledge travel funding from the University of Western Australia.

## Author contributions

H.M. conceived the concept. All authors planned the experiments. T.J.J.I. and K.T.M. selected bacterial isolates for the MIC experiments, and T.F.P. prepared the isolates and ran the broth microdilutions. J.M.S provided data on antibiotic susceptibility and resistance mechanisms of strains used and prepared fresh overnight plates for analysis. D.C.S. created the experimental setup and ran the micro-cytometer experiments. D.C.S. wrote the analysis software and analysed the data. H.M. and D.C.S. prepared the manuscript. H.M., D.C.S., J.M.S., T.F.P. and T.J.J.I. reviewed and contributed to the manuscript.

## Competing interests

H.M., D.C.S., and T.J.J.I are authors on patent application WO 2020/058682. H.M. and D.C.S. are authors on patent application WO 2020/058681 A1. T.F.P., K.T.M. and J.M.S. declare no competing interests.
