## [Peer Review File · Nature Communications]

Reviewers' comments:

Reviewer #1 (Remarks to the Author):

In the submitted report Spencer and colleagues report on the development of microfluid impedance cytometry to obtain phenotypic antimicrobial susceptibility results in less than 1 hour – iFAST. The authors conclude that their assay shows excellent concordance against classical broth microdilution for a range of antibiotics and bacterial species. Technically speaking, iFAST is based on previous cultural isolation of the pathogen which itself requires 16-24 hrs. Thus, the total time from obtaining the sample to AST result is largely determined by the time required for cultural isolation of the pathogen.

I am intrigued by the procedure developed, however, the data produced are somewhat preliminary – at least from a microbiological perspective. As is, the report is basically a technical report, which points to the exciting possibility of using microfluid impedance cytometry for antimicrobial susceptibility testing. It is legitimate that the authors have focused on a limited number of bacterial species and defined antibiotics, but I would strongly encourage the authors to study a larger number of clinical isolates ($n > 100$) of a given species to validate and support their exciting development on a broader level.

The comparison of iFAST to the gold standard broth microdilution should be done at the MIC level using a range of clinical isolates covering all possible MICs. Comparison should be done by a systematic side by side comparison, e.g. in an x/y format which allows visualization of > 100 data points, where x is the quantitative MIC value for iFAST and y the quantitative MIC value for broth microdilution – with the view to quantify the concordance of iFAST with that of broth microdilution and to assess the possible CBP categorization errors. Please note that CBP data are derived data – derived from MIC analysis. Thus, first MIC determination is necessary, followed by CBP categorization.

Additional comments:

1. Please indicate whether the MRSA strain is an inducible or constitutive MecaA-expressing strain. A fair number of clinical MRSA strains should be studied to cover both genotypes.

2. Line 30/31: "Nearly all antibiotic testing is currently performed using classical culture-dependent microbiology methods that provide a susceptibility profile within 24 to 48 hours, or longer." Please note that rapid and fully automated culture-based AST methods have been developed, which require not more than 6 hrs, e.g. JAC 2017, 72: 3063-3069.

Reviewer
Erik C. Böttger

Reviewer #2 (Remarks to the Author):

Due to the urgent need for a quick test that provides information on prescribing appropriate antibiotics and dosages, the authors introduce their impedance-based Fast Antimicrobial Susceptibility Test (iFAST). The iFAST assay is used for rapid detection of antibiotic susceptibility for clinically relevant bacterial cells and different pathogen-antibiotics combinations. The iFAST device is a label-free assay that utilizes a collection of single-cell data of $\sim 10k+$ bacteria cells that can be collected in 2-3 minutes. The authors also describe a method for increasing sensitivity of the standard microfluidic impedance cytometry methods via a combination of two differential

measurements. The assay obtains complex impedance information which can be used to calculate the opacity and electrical diameter, which is then used to evaluate different antibiotic dosages vs control for different bacteria cell populations. The simple and rapid assay has shown similar statistically relevant results when compared to standard benchmark assays for a variety of antibiotics and bacterial species.

While the paper is significant, there are concerns and issues that need to be addressed as detailed below:

- Using electrical properties for separation or identification of bacteria has been performed by other researchers. The authors should compare the novelty of their method with dielectrophoretic techniques, microelectrode arrays, and other methods relying on biophysical properties. Papers published by researchers at RTI, Georgia Tech, Virginia Tech, and others can be used for comparison. The authors should also discuss the superiority of their technique with other rapid antibiotic susceptibility tests such as (<https://www.pnas.org/content/114/34/9170>) and (<https://mbio.asm.org/content/11/1/e03109-19>).
- Can the authors explain the reason why the rapid AST needs to be "ideally done within an hour." Would 2hrs, 4hrs, etc. be sufficient for clinical applications?
- Does the claim to "deliver result in less than 1 hour" include incubation, antibiotic exposure, sample preparation, experimentation and data analysis times? Authors mention incubation for 30min, antibiotic exposure for 30min and data collection for ~3min.
- What is the reason behind analyzing 10^5 cells at single-cell level? How it is ensured that only once cell passes through the electrodes at a given time. How do you prevent multiple cells passing through the sensing region at once? Is there any data filtration for non-single cells that transit over the electrodes?
- Why the authors define opacity as 40MHz/5MHz? What is the significance of these frequencies? Is it possible that other frequencies have better results?
- Were the same devices used for the different cell populations and experiments? How authors address device-to-device variations?
- In Figure 1B, it looks like electrical diameter is enough to distinguish between the exposed/unexposed cell populations. Why do the authors need to include Opacity?
- Does the time between ending the exposure to antibiotics (like washing the cells with HBSS) and experimentation with iFAST device matter? For example, conducting iFAST assay experimentation immediately after washing/suspending cells in HBSS vs waiting an additional 15-30min after washing/suspending cells before experimentation.
- In ESI Fig S2, should the x-axis label be frequencies ranging from 4 Hz to 10 Hz?

Reviewer #3 (Remarks to the Author):

This paper by Daniel C. Spencer et al presents a new method for determining bacteria antibiotic susceptibility within 30 min. The data is technically sound and the results are novel. The conclusions presented in the paper are well founded, although the reviewer finds that some of them need to be further substantiated. Moreover, in order for the manuscript to be useful to scientists in this field, further data need to be made available (see specific comments for more

details). Therefore I find that the authors need to revise the manuscript to address specific concerns before a final decision is reached.

More specifically this paper presents an impedance based method to determine the susceptibility of several bacteria strains to antibiotics. The authors claim that their method is faster than any other standard, electrochemical or flow cytometry based method, giving a result in 30 minutes, which is as good as the golden standard. The claims are novel, especially the device that the authors are using to record the data, which indeed presents a big improvement compared to other impedance based systems.

However, I find that particularly the lack of more information regarding the experimental setup is a serious disadvantage of the paper, since this is where the novelty lies. Currently the setup is only briefly presented in the supplementary information (figure S1b in particular). With the standard knowledge of impedance based methods it is very difficult for the reader to understand the reason why this system has such a better limit of detection from the information presented in the article. I think figure S1b is rather important to have in the main text, along with a more thorough explanation of why it improves SNR so much and allows for such a large channel and good signal for such small particles. This is one of the main findings of the paper!

I further find that the claim of 30 min time to result is not fully explained. The methods described in the paper all take much longer than 30 min, some have overnight culturing. I assume that this is just for the experimental strains, but it is not very clear in the text how a possible real-life experiment would work in about 30 minutes. Also in terms of what type of sample handling would be required.

The results are discussed in the context of previous literature, but the presentation is a bit clumsy. My first reaction was that this part is something for the introduction and not for the discussion. Some rephrasing may be in order here.

Furthermore, although the manuscript is quite well written, there are several typographical and sentence building errors that in a few cases actually impede the understanding of the paper. Particularly in the discussion part there are references to the wrong figures in the supplementary section, in the data analysis section there is a first mention to "beads", which has never been used before in the text so the reader cannot see what these are used for and why, and it is not very clear what the authors mean by "gate".

The manuscript should not be shortened; I actually think that more information should be added in order to communicate the findings, as mentioned above. Reproducing the experiments would be difficult, given the lack of details regarding the setup. Here I find it essential that the authors provide further information regarding the fabrication (e.g. details on bonding, details on connections, e.g. brand, producer etc), amplification of the signal, etc.

The authors have adequately addressed existing literature, both in terms of current standard methods, but also in terms of other less conventional methods. Statistical analysis has been conducted.

Below I list all the specific comments in the manuscript.

Line 148. One of the main finding in this paper is the improvement of SRN in large micro channels for such small particles. Therefor I believe it is important to have figure S1b in the main text, together with a more thorough explanation.

Line 168 to 169, small error in text, the sentence starting with. Antibiotic-induced... should be rewritten.

Line 174, the parentheses is misplaced.

Ling 179: I get confused in the description here. The time to results is stated to be 30 min.

However described here is two steps of 30 min., also in figure 1C there is an overnight culturing step. An explanation of how this will be applicable in real life application would be useful.

Line 196, The data is plotted as the % of cells in a gate. A definition of "gate" is lacking.

Line 343: The sentence in terms of... should be rephrased it makes no sense.

Line 369: A small comment: I find this comparison of dyes better placed in the introduction and not in the discussion of own results.

Line 390 and 405, I think the figure reference is wrong, it should be S4 instead of S1.

Line 420, delete "in"

Line 433, I find it hard to figure out what is meant by "pre--defined gate" and why data analysis on beads are discussed. Please elaborate.

Small correction,

line 316 (d is missing in use, should be used.)

line 317 (a to is missing in the sentence "due to its"

line 321 (the sentence should read " a similar trend following exposure to Colistin is observed"

Reviewer #1 (Remarks to the Author):

In the submitted report Spencer and colleagues report on the development of microfluid impedance cytometry to obtain phenotypic antimicrobial susceptibility results in less than 1 hour - iFAST. The authors conclude that their assay shows excellent concordance against classical broth microdilution for a range of antibiotics and bacterial species. Technically speaking, iFAST is based on previous cultural isolation of the pathogen which itself requires 16-24 hrs. Thus, the total time from obtaining the sample to AST result is largely determined by the time required for cultural isolation of the pathogen.

The reviewer is absolutely correct and this is true for all current AST. For example, broth micro-dilution requires culture isolation with a further 18hours before test result. We discussed this restriction in the paper but we have now included further clarification.

I am intrigued by the procedure developed, however, the data produced are somewhat preliminary - at least from a microbiological perspective. As is, the report is basically a technical report, which points to the exciting possibility of using microfluid impedance cytometry for antimicrobial susceptibility testing. It is legitimate that the authors have focused on a limited number of bacterial species and defined antibiotics, but I would strongly encourage the authors to study a larger number of clinical isolates (n>100) of a given species to validate and support their exciting development on a broader level.

We thank the reviewer for their positive comments regarding the potential application of the method in the field of antimicrobial susceptibility testing. The focus of this manuscript is indeed the technical aspect of the method, which integrates together diverse aspects of microfluidics and electronics to deliver an innovative platform that addresses a very important microbiology challenge. The method has shown remarkable utility with a wide range of antibiotics and with high priority drug-resistance pathogen resulting in intriguing mechanism-specific signatures related to antimicrobial effects.

We agree with the reviewer of the need to study a large number of clinical isolates. We have recently secured funding for a large project that performs extensive validation of the technology in relevant laboratory and clinical settings. This validation study addresses a different phase of the work. Unfortunately, the Covid pandemic means that this project will not start until 2021. We believe that the submitted manuscript reports an important new approach to AMR diagnostics that needs to be made available to the scientific community more rapidly than current timelines will allow, particularly in view of the urgent and unmet challenges in this area.

The comparison of iFAST to the gold standard broth microdilution should be done at the MIC level using a range of clinical isolates covering all possible MICs. Comparison should be done by a systematic side by side comparison, e.g. in an x/y format which allows visualization of >100 data points, where x is the quantitative MIC value for iFAST and y the quantitative MIC value for broth microdilution - with the view to quantify the concordance of iFAST with that of broth microdilution and to assess the possible CBP categorization errors. Please note that CBP data are derived data - derived from MIC analysis. Thus, first MIC determination is necessary, followed by CBP categorization.

We thank the reviewer for their careful consideration of this point. We are intending to undertake a comprehensive comparison of iFAST to the gold standard broth microdilution

over the next 2 years (see above). This will be done at the MIC level using many different clinical isolates and patient samples that cover many different MICs. We would highlight that the data sets in our paper shows a remarkable degree of concordance between classical MICs and impedance-based MIC in terms of resistance/susceptibility, even more so given that the comparison is between a 20 hour MIC measurement and an approximately 1 hour impedance test. The issue of concordance between classical MIC and rapid test MICs is recognised by EUCAST, and the use and validation of rapid susceptibility tests remains an area that requires further research in order to understand equivalence and relate this to the derived CBP definitions. This extensive programme of work falls outside the scope of the current manuscript.

Additional comments:

1. Please indicate whether the MRSA strain is an inducible or constitutive MecA-expressing strain. A fair number of clinical MRSA strains should be studied to cover both genotypes.

The MRSA strain is a constitutive MecA expressing strain. The isolate is a representative of one of the predominant epidemic lineages of MRSA (EMRSA 15) and represents many strains found in the clinic. We intend to study other strains with inducible resistance as part of the laboratory clinical evaluation study which is postponed until 2021. This point has been added in the revised manuscript and in ESI Table 1.

2. Line 30/31: "Nearly all antibiotic testing is currently performed using classical culture-dependent microbiology methods that provide a susceptibility profile within 24 to 48 hours, or longer." Please note that rapid and fully automated culture-based AST methods have been developed, which require not more than 6 hrs, e.g. JAC 2017, 72: 3063-3069.

We agree that such platforms are available, but following several discussions with a range of microbiology labs in UK hospitals it appears that these are not that widely used. We have modified the paper to reflect this.

Reviewer #2 (Remarks to the Author):

Due to the urgent need for a quick test that provides information on prescribing appropriate antibiotics and dosages, the authors introduce their impedance-based Fast Antimicrobial Susceptibility Test (iFAST). The iFAST assay is used for rapid detection of antibiotic susceptibility for clinically relevant bacterial cells and different pathogen-antibiotics combinations. The iFAST device is a label-free assay that utilizes a collection of single-cell data of ~10k+ bacteria cells that can be collected in 2-3 minutes. The authors also describe a method for increasing sensitivity of the standard microfluidic impedance cytometry methods via a combination of two differential measurements. The assay obtains complex impedance information which can be used to calculate the opacity and electrical diameter, which is then used to evaluate different antibiotic dosages vs control for different bacteria cell populations. The simple and rapid assay has shown similar statistically relevant results when compared to standard benchmark assays for a variety of antibiotics and bacterial species.

While the paper is significant, there are concerns and issues that need to be addressed as detailed below:

- Using electrical properties for separation or identification of bacteria has been performed by other researchers. The authors should compare the novelty of their method with dielectrophoretic techniques, microelectrode arrays, and other methods relying on biophysical properties. Papers published by researchers at RTI, Georgia Tech, Virginia Tech, and others can be used for comparison. The authors should also discuss the superiority of their technique with other rapid antibiotic susceptibility tests such as

(<https://eur03.safelinks.protection.outlook.com/?url=https%3A%2F%2Fwww.pnas.org%2Fcontent%2F114%2F34%2F9170&data=01%7C01%7Chm%40ecs.soton.ac.uk%7Cd3f872a0cd564c0a92d008d7e10b1596%7C4a5378f929f44d3e8e89669d03ada9d8%7C0&sd=0&reserved=0>) and

(<https://eur03.safelinks.protection.outlook.com/?url=https%3A%2F%2Fmbio.asm.org%2Fcontent%2F11%2F1%2Fe03109-19&data=01%7C01%7Chm%40ecs.soton.ac.uk%7Cd3f872a0cd564c0a92d008d7e10b1596%7C4a5378f929f44d3e8e89669d03ada9d8%7C0&sd=0&reserved=0>) and (<https://eur03.safelinks.protection.outlook.com/?url=https%3A%2F%2Fmbio.asm.org%2Fcontent%2F11%2F1%2Fe03109-19&data=01%7C01%7Chm%40ecs.soton.ac.uk%7Cd3f872a0cd564c0a92d008d7e10b1596%7C4a5378f929f44d3e8e89669d03ada9d8%7C0&sd=0&reserved=0>) and (<https://eur03.safelinks.protection.outlook.com/?url=https%3A%2F%2Fmbio.asm.org%2Fcontent%2F11%2F1%2Fe03109-19&data=01%7C01%7Chm%40ecs.soton.ac.uk%7Cd3f872a0cd564c0a92d008d7e10b1596%7C4a5378f929f44d3e8e89669d03ada9d8%7C0&sd=0&reserved=0>).

We thank the reviewer for identifying further groups working in this field. We have in fact cited and discussed work from Georgia Tech at length (see reference 12). We have included references to the additional papers suggested by the reviewer and extended the manuscript to draw comparison with other techniques

The following additional relevant papers have been cited:

- Sensors, 18(10), 3496, October 2018
- Lab on a Chip, 14(13), 2327-2333, 2014
- Journal of Microelectromechanical Systems, 27(5), 810-817, October 2018
- PNAS 114(34) 9170-9175 2017
- Wistrand-Yuen P, Malmberg C, Fatsis- Kavalopoulos N, Lübke M, Tängdén T, Kreuger J. 2020. A multiplex fluidic chip for rapid phenotypic antibiotic susceptibility testing. mBio 11:e03109-19. <https://doi.org/10.1128/mBio.03109-19>.
- Choi et al A rapid antimicrobial susceptibility test based on single-cell morphological analysis Science Translational Medicine 17 Dec 2014: Vol. 6, Issue 267, pp. 267ra174 DOI: 10.1126/scitranslmed.3009650

Concerning RTI; if we understand the reviewer correctly, RTI refers to projects funded by the CARB-X organisation. We have identified 5 diagnostic companies that are working in this space, but none have scientific publications as far as we are aware.

- Tallis (genomic testing) and DayZero Diagnostics (whole genome sequencing and machine learning).
- Proteus – optical fibre based imaging of lungs using smart fluorescent probes
- Pattern Biosciences (Klaris) – combines AI with single cell imaging for fast phenotypic test
- HelixBind – sepsis test for 20 different microorganisms with proprietary sample purification process.

The market lead in rapid diagnosis is Accelerate Pheno and we have now included a sentence to highlight this company's impact.

- Can the authors explain the reason why the rapid AST needs to be "ideally done within an hour." Would 2hrs, 4hrs, etc. be sufficient for clinical applications?

This point and have qualified in the text as follows:

"A rapid AST that provides rapid turnaround and data within a a shift day would have a major impact on many clinical applications. A much-reduced time to result (e.g. around 1 hour post-culture) would be particularly advantageous in providing information promptly enabling clinicians to expedite evidence-based prescribing"

- Does the claim to "deliver result in less than 1 hour" include incubation, antibiotic exposure, sample preparation, experimentation and data analysis times? Authors mention incubation for 30min, antibiotic exposure for 30min and data collection for ~3min.

The reviewer is correct – the total time including all preparation is approximately 1 hour. We have clarified this point in the text. The preincubation time could be reduced (or even eliminated in some cases, but this needs further work), and the exposure could be dropped to 15 to 20 mins (c.f. ESI Fig S3). The analysis is quick and is automated. Therefore for the sake of simplicity we stat that the test takes approximately 1 hour (or less).

- What is the reason behind analyzing 10^5 cells at single-cell level?

10^5 cells were chosen as the target number for analysis in order to have a statistically meaningful sample number. Furthermore, this number of cells aligns with standard MIC methods (as defined by EUCAST/CLSI with a 5×10^5 titre).

- How it is ensured that only once cell passes through the electrodes at a given time. How do you prevent multiple cells passing through the sensing region at once? Is there any data filtration for non-single cells that transit over the electrodes?

More than one cell can pass through the measurement region (coincidence) and the probability of this occurring is defined by Poisson statistics. There is a well-known trade-off for cytometric analysis method between throughput and coincidence. In our experiment cells are diluted to a concentration where the probability of more than one cell passing through the

measurement volume is extremely low. Furthermore, non-single cells will have a very different impedance signal profile which is removed using gating.

- Why the authors define opacity as 40MHz/5MHz? What is the significance of these frequencies? Is it possible that other frequencies have better results?

These two frequencies are used to characterise the size (electrical diameter at 5MHz), and the cell wall properties (at 40MHz). This is explained in Fig 2 ESI and the figure legend (update) to Figure 1 main text

- Were the same devices used for the different cell populations and experiments? How authors address device-to-device variations?

Device to device variability is entirely eliminated by the use of polystyrene beads as reference particles. This has been explained in a recent article of ours (Morgan and Spencer ACS Sensors ACS Sens. 5(2) 423-430 2020). We have now included a section under "Measurement Principles" to explain this.

- In Figure 1B, it looks like electrical diameter is enough to distinguish between the exposed/unexposed cell populations. Why do the authors need to include Opacity?

The reviewer is correct in that in some cases the changes from antibiotic exposure are apparent just in the electrical diameter. However opacity provides additional information for some classes of antimicrobial agent. Opacity provides important information on the cell wall and cell membrane properties which are intimately linked to the mode of action of certain antibiotics. Furthermore, the gating of populations (from noise, beads and solid debris) is made much simpler using opacity data.

- Does the time between ending the exposure to antibiotics (like washing the cells with HBSS) and experimentation with iFAST device matter? For example, conducting iFAST assay experimentation immediately after washing/suspending cells in HBSS vs waiting an additional 15-30min after washing/suspending cells before experimentation.

ESI Fig S3 shows how response changes with time after a 30 minute incubation with antibiotic; these results led to the choice of 30 minutes antibiotic exposure. We have not comprehensively measured the response for different wait times after incubation with antibiotic. Cells re-suspended in HBSS (without antibiotic) will eventually recover and begin to grow, depending on the concentration and class of antibiotic. This effect will eventually be systematically studied but there are a large number of variables. To ensure consistency and a reliable test, cells were measured as soon as possible (< 10minutes) after 30minutes incubation.

- In ESI Fig S2, should the x-axis label be frequencies ranging from 4 Hz to 10 Hz?

We thank the reviewer for pointing out this error; it should be a \log_{10} scale – corrected.

Reviewer #3 (Remarks to the Author):

This paper by Daniel C. Spencer et al presents a new method for determining bacteria antibiotic susceptibility within 30 min. The data is technically sound and the results are

novel. The conclusions presented in the paper are well founded, although the reviewer finds that some of them need to be further substantiated. Moreover, in order for the manuscript to be useful to scientists in this field, further data need to be made available (see specific comments for more details). Therefore I find that the authors need to revise the manuscript to address specific concerns before a final decision is reached.

More specifically this paper presents an impedance based method to determine the susceptibility of several bacteria strains to antibiotics. The authors claim that their method is faster than any other standard, electrochemical or flow cytometry based method, giving a result in 30 minutes, which is as good as the golden standard. The claims are novel, especially the device that the authors are using to record the data, which indeed presents a big improvement compared to other impedance based systems.

However, I find that particularly the lack of more information regarding the experimental setup is a serious disadvantage of the paper, since this is where the novelty lies. Currently the setup is only briefly presented in the supplementary information (figure S1b in particular). With the standard knowledge of impedance based methods it is very difficult for the reader to understand the reason why this system has such a better limit of detection from the information presented in the article. I think figure S1b is rather important to have in the main text, along with a more thorough explanation of why it improves SNR so much and allows for such a large channel and good signal for such small particles. This is one of the main findings of the paper!

We thank the reviewer for their observations. The aim was to ensure the paper appealed to the widest possible audience and for this reason we kept the explanation of the technical innovations succinct in the main text, with further details for the interested reader in ESI. We have revised the paper to include more details of this experimental innovation – paragraph 2 of measurement principles.

I further find that the claim of 30 min time to result is not fully explained. The methods described in the paper all take much longer than 30 min, some have overnight culturing. I assume that this is just for the experimental strains, but it is not very clear in the text how a possible real-life experiment would work in about 30 minutes. Also in terms of what type of sample handling would be required.

The standard protocols used in nearly all testing labs requires an overnight culture prior to an AST. Generally a sample is taken, e.g. of urine or blood and the bacteria in this sample grown either on a purity (agar) plate or in a blood bottle to produce sufficient numbers for subsequent analysis. In principle, our test could be done directly from sample (e.g. urine for UTI) but our aim was to develop a rapid test that conforms to standard clinical workflow. The test should seamlessly integrate with current microbiology practice and workflow (e.g. in hospitals), and rapidly accelerate the time to answer (see Fig 1c).

The antibiotic incubation time is indeed 30 minutes, with a 3 minute measurement time. In this work, we included an additional pre-incubation step of 30 minutes, which indeed makes the overall time approximately 1 hour. This pre-incubation time was required because we used clinical isolates. It may not be needed in practice and we intend to optimise these times as the test is developed for clinical utilisation. We have amended the paper to make clear the 1 hour test time.

For a typical lab based protocol the sample handling could be done robotically and is minimal (as per Fig 1 and current ASTs).

The results are discussed in the context of previous literature, but the presentation is a bit clumsy. My first reaction was that this part is something for the introduction and not for the discussion. Some rephrasing may be in order here.

Some of this text was originally in the introduction we decided that it would be better in the discussion section, specifically where we refer to a comparison with optical flow cytometry which has been proposed as an alternative method for many years. It is important to draw comparisons between the two different cytometric methods which was our aim since many biologists are familiar with optical flow cytometry but may not be familiar with the electrical analogue. We have taken the reviewers advice and re-phrased the discussion section.

Furthermore, although the manuscript is quite well written, there are several typographical and sentence building errors that in a few cases actually impede the understanding of the paper. Particularly in the discussion part there are references to the wrong figures in the supplementary section, in the data analysis section there is a first mention to "beads", which has never been used before in the text so the reader cannot see what these are used for and why, and it is not very clear what the authors mean by "gate".

Polystyrene beads are mentioned in the Methods section but the reviewer may have missed this. We have now made it clear what the purpose of the beads are and cited a new paper (also see answer to reviewer 2).

"Gate" is a standard term used in flow cytometry to define a population of cells on a scatter plot. We have used the word "contour" and "gate" in the revised manuscript

The manuscript should not be shortened; I actually think that more information should be added in order to communicate the findings, as mentioned above. Reproducing the experiments would be difficult, given the lack of details regarding the setup. Here I find it essential that the authors provide further information regarding the fabrication (e.g. details on bonding, details on connections, e.g. brand, producer etc), amplification of the signal, etc.

The authors have adequately addressed existing literature, both in terms of current standard methods, but also in terms of other less conventional methods. Statistical analysis has been conducted.

Following on from this recommendation, we have added further information in the main text and in the methods section (specifically in "Impedance Cytometer").

Below I list all the specific comments in the manuscript.

Line 148. One of the main finding in this paper is the improvement of SRN in large micro channels for such small particles. Therefor I believe it is important to have figure S1b in the main text, together with a more thorough explanation.

We have improved the explanation as to how the novel system works in the paper – see “Measurement Principle”. Fig 1(a) is very similar to ESI Fig S1 but has less detail. It is important the paper can be appreciated by a general readership, which is why the specific details on the electronics are in the ESI

Line 168 to 169, small error in text, the sentence starting with. Antibiotic-induced... should be rewritten.

Done

Line 174, the parentheses is misplaced.

Done

Line 179: I get confused in the description here. The time to results is stated to be 30 min. However described here is two steps of 30 min., also in figure 1C there is an overnight culturing step. An explanation of how this will be applicable in real life application would be useful.

We apologise for the confusion. The statement is amended to clarify that it is 30 minutes from addition of antibiotics. We have also clarified that this procedure reflects the current workflow used in many hospitals, in the UK, Australia and worldwide, where susceptibility testing follows an overnight culture phase. We have amended the text to clarify this

Line 196, The data is plotted as the % of cells in a gate. A definition of "gate" is lacking.

See above – but have used the term contour as well as gate

Line 343: The sentence in terms of... should be rephrased it makes no sense.

Done

Line 369: A small comment: I find this comparison of dyes better placed in the introduction and not in the discussion of own results.

We edited the section in the discussion

Line 390 and 405, I think the figure reference is wrong, it should be S4 instead of S1.

Apologies – now corrected

Line 420, delete "in"

Done

Line 433, I find it hard to figure out what is meant by "pre-defined gate" and why data analysis on beads are discussed. Please elaborate.

Defining gate helps with understanding a pre-defined gate. Text has been edited.

Small correction,

line 316 (d is missing in use, should be used.)

line 317 (a to is missing in the sentence "due to its"

line 321 the sentence should read " a similar trend following exposure to Colistin is observed"

Done – and we thank the reviewer for their very careful reading of the manuscript!

REVIEWERS' COMMENTS:

Reviewer #2 (Remarks to the Author):

The authors have addressed the concerns of the reviewers. However, still remains the issue of calling it FAST. The detection is fast but the process is not fast. I am wondering why the authors remove the sample preparation time out of the equation.

Reviewer #2 (Remarks to the Author):

The authors have addressed the concerns of the reviewers. However, still remains the issue of calling it FAST. The detection is fast but the process is not fast. I am wondering why the authors remove the sample preparation time out of the equation.

Thank you for your comment. We have changed the title of the article to remove the term "iFAST". We have modified the text again to make it very clear that the bacteria are taken from an overnight culture so that the test mirrors standard microbiological ASTs. The actual time for the test is 30 minutes incubation followed by 3 minutes for measurement. Therefore, we respectfully maintain that the test is indeed fast when compared with current ASTs that take many hours after the sample preparation time.